# Performance and Outcomes of Routine Viral Load Testing in People Living with HIV Newly Initiating ART in the Integrated HIV Care Program in Myanmar between January 2016 and December 2017

**DOI:** 10.3390/tropicalmed5030140

**Published:** 2020-08-31

**Authors:** Sai Soe Thu Ya, Anthony D. Harries, Khin Thet Wai, Nang Thu Thu Kyaw, Thet Ko Aung, July Moe, Thurain Htun, Htet Naing Shin, Mar Mar Aye, Htun Nyunt Oo

**Affiliations:** 1Center for Operational Research, International Union Against Tuberculosis and Lung Disease, Myanmar Office, Mandalay 05021, Myanmar; nangthu82@gmail.com (N.T.T.K.); dr.thetko@gmail.com (T.K.A.); julymoe.dr@gmail.com (J.M.); drthurain07@gmail.com (T.H.); htetnaingshin01@gmail.com (H.N.S.); 2Center for Operational Research, International Union Against Tuberculosis and Lung Disease, 75006 Paris, France; adharries@theunion.org; 3Department of Infectious and Tropical Diseases, London School of Hygiene and Tropical Medicine, London WC1E 7HT, UK; 4Department of Medical Research, Ministry of Health and Sports, Yangon 11191, Myanmar; khinthetwaidmr@gmail.com; 5Department of Medical Science, MGH, Ministry of Health and Sports, Nay Pyi Taw 15012, Myanmar; marmaraye.mu1@gmail.com; 6National AIDS Program, Ministry of Health and Sports, Nay Pyi Taw 15012, Myanmar; tunnyuntoo13@gmail.com

**Keywords:** antiretroviral therapy, virologic failure, first-line antiretroviral therapy, second-line antiretroviral therapy, SORT IT

## Abstract

Myanmar has introduced routine viral load (VL) testing for people living with HIV (PLHIV) starting first-line antiretroviral therapy (ART). The first VL test was initially scheduled at 12-months and one year later this changed to 6-months. Using routinely collected secondary data, we assessed program performance of routine VL testing at 12-months and 6-months in PLHIV starting ART in the Integrated HIV-Care Program, Myanmar, from January 2016 to December 2017. There were 7153 PLHIV scheduled for VL testing at 12-months and 1976 scheduled for VL testing at 6-months. Among those eligible for testing, the first VL test was performed in 3476 (51%) of the 12-month cohort and 952 (50%) of the 6-month cohort. In the 12-month cohort, 10% had VL > 1000 copies/mL, 79% had repeat VL tests, 42% had repeat VL > 1000 copies/mL (virologic failure) and 85% were switched to second-line ART. In the 6-month cohort, 11% had VL > 1000 copies/mL, 83% had repeat VL tests, 26% had repeat VL > 1000 copies/mL (virologic failure) and 39% were switched to second-line ART. In conclusion, half of PLHIV initiated on ART had VL testing as scheduled at 12-months or 6-months, but fewer PLHIV in the 6-month cohort were diagnosed with virologic failure and switched to second-line ART. Programmatic implications are discussed.

## 1. Introduction

The scale up of antiretroviral therapy (ART) has been one of the great public health success stories of our time. By June 2019, there were 24.5 million people globally receiving ART, representing 62% of all people living with HIV (PLHIV) [1]. Treatment not only benefits the HIV-infected individual by reducing morbidity and mortality, but also significantly decreases the risk of HIV transmission to non-infected partners. The success of ART scale up led the Joint United Nations Program on HIV/AIDS (UNAIDS) to release its 90-90-90 treatment targets for HIV [2]. These targets specify that by 2020, 90% of individuals living with HIV will know their HIV status, 90% of people with diagnosed HIV infection will receive sustained ART and 90% of those on ART will be virally suppressed. Modeling studies indicate that meeting these 90-90-90 targets will enable the world to end the AIDS epidemic by 2030, defined as a 90% reduction in both incidence of HIV and AIDS-related mortality [2]. Based on the excellent global progress made to date and to quicken the pace of implementation, UNAIDS published fast-track treatment targets (now 95-95-95 instead of 90-90-90) in a supreme effort to bring the AIDS epidemic to an end by 2030 [3].

The response to ART in the initial years of treatment scale-up was monitored clinically and by CD4 cell count. However, the deficiencies of these methods, particularly in the high HIV-burden countries in sub-Saharan Africa [4], led to the adoption of routine viral load (VL) measurement as the gold standard for assessing response to treatment, and this is now the preferred monitoring approach to diagnose and confirm ART failure [5,6]. Guidance from the World Health Organization (WHO) in 2017 specified that VL testing should be conducted at 6-months and 12-months after ART initiation and every 12-months thereafter [7]. A VL > 1000 copies/mL in the repeat VL test after adherence counseling is used to determine treatment failure and the need to change from first-line to second-line ART. VL testing is important because primary and acquired HIV drug resistance and ART failure is a growing threat globally to epidemic control [8,9], and if this is not diagnosed early and managed properly there is further risk of resistance amplification and non-response to ART regimens.

In Myanmar, the HIV epidemic is concentrated among key populations and in 2018 there were an estimated 240,000 people living with HIV (PLHIV) of whom about 167,000 (70%) were on ART [10,11]. In 2018, about 54% of PLHIV on ART received VL testing in the previous 12 months of whom 92% had a suppressed VL result [10,11]. Initially, VL monitoring was targeted for PLHIV who were suspected of treatment failure on clinical or immunological criteria. However, national guidelines were changed in 2017 and recommended that routine VL testing be done in PLHIV on ART in line with WHO recommendations [12]. While routine VL is now an accepted practice, its cost and complexity and the need to ensure that results are acted upon has led some people to question its sustainability and cost-effectiveness especially in resource-poor settings [13,14].

The International Union Against Tuberculosis and Lung Disease (The Union) has been implementing an integrated HIV care (IHC) program in Myanmar in collaboration with the national AIDS program (NAP) since 2005. The program introduced routine VL testing in January 2017. Initially, the first VL test was done at 12-months after starting ART. One year later, in January 2018, the policy changed to do the first VL test at 6-months after starting ART. Although this change has already taken place, there is no information about whether health workers have been compliant with policy or whether VL testing at 6-months identifies the same proportion of PLHIV with high VL and ART failure as compared with VL testing at 12-months. For example, it is possible in PLHIV with high baseline VL and advanced HIV-related disease that viral suppression has still not occurred by 6-months and such patients with VL > 1000 copies/mL might mistakenly be diagnosed as failing treatment. Alternatively, true virologic failures may be more common at 12-months rather than 6-months. The change in policy and practice in Myanmar between 2017 and 2018 provided an opportunity to describe and assess programmatic performance and outcomes of doing the first routine VL testing at 12-months and at 6-months.

The aim of this study therefore was to describe and report on the programmatic performance and outcomes of routine VL testing at 12-months and at 6-months in PLHIV newly initiating first-line ART in the IHC Program in Myanmar. Specific objectives for PLHIV in the two cohorts, those VL tested at 12-months or those VL tested at 6-months, were to (i) describe and compare baseline characteristics, (ii) document who had VL testing, both within and outside of the scheduled times, (iii) describe the cascade and outcomes of VL testing including the diagnosis of virologic failure and switch to second-line ART and iv) assess risk factors for virologic failure amongst those who were VL tested within the scheduled time periods.

## 2. Materials and Methods

### 2.1. Study Design

This was a retrospective cohort study using routinely collected secondary data.

### 2.2. Setting

#### 2.2.1. General Setting

The Republic of the Union of Myanmar, with a total surface area of 676,578 km^2^, is located in mainland Southeast Asia and it borders China to the northeast, India and Bangladesh to the west, Laos to east and Thailand to the east and southeast. According to the provisional results of the 2014 Census, Myanmar had a total population of about 52 million [15]. Myanmar is one of the least developed countries in the region with a GDP per capita of USD$1264 in 2017 [16]. The country is divided into 7 states and 7 regions and the Union Territory Region, Nay Pyi Taw, and it is administratively subdivided into districts, townships, wards and villages. Seventy percent of the population lives in rural regions.

#### 2.2.2. Integrated HIV Care (IHC) clinics and VL Testing

The Union has been implementing an “Integrated HIV Care Program IHC” in collaboration with the National AIDS Program (NAP) and National Tuberculosis Program (NTP) in Mandalay Region, Sagaing Region, Magway Region, Yangon Region and Shan State since 2005. The details of the NAP and Union IHC program have been previously described [17].

In 2018, there were 49 IHC clinics in 37 townships in Myanmar. Every month around 370 patients with a confirmed HIV diagnosis were registered for care and treatment at Union IHC clinics. After enrolment, all PLHIV were evaluated for ART eligibility and started ART if they were eligible according to the most recent national ART guidelines (which currently recommend the “HIV test and treat” approach in line with WHO guidance) [6]. There were around 320 PLHIV who started ART every month in the Union IHC clinics. Follow-up visits occurred at 2–4 weeks after initial enrolment and then 3 monthly and 6 monthly depending on the clinical condition. At every visit, PLHIV were monitored for side effects, complications and drug interactions. Additional visits were requested whenever they were needed.

Before 2017, PLHIV started on ART were monitored by CD4 cell counts and clinical assessment to decide on whether they were failing ART. Those who were judged to have failed treatment on those grounds were targeted for VL testing—PLHIV with VL > 1000 copies/mL received adherence counseling and if the repeat test still showed VL > 1000 copies/mL, a diagnosis of virologic failure was made and first-line treatment was changed to second-line treatment. In January 2017, the program introduced routine VL testing, with the first test done at 12-months after starting ART. One year later, in January 2018, the policy changed to do the first VL test at 6-months after starting ART. In brief, the procedure of VL testing involved the collection of venous or capillary whole blood or plasma at the public laboratories where IHC clinics implemented activities, with specimens then transported to the central Public Health Laboratory, Mandalay. The VL assay was performed at the central laboratory and results sent back to the clinics.

The management of patients was based on the results of VL testing. In brief, if the first VL > 1000 copies/mL, enhanced adherence counseling (EAC) took place, which involved one intensive counseling session by social workers under the guidance of medical officers. A repeat VL test was performed 3–6 months later. If VL < 1000 copies/mL the patient was maintained on first-line ART. If VL > 1000 copies/mL, a diagnosis of virologic failure was made and the patient was switched to second-line therapy.

Every patient registered in the IHC program had a dedicated file in which baseline and follow-up clinical details and laboratory test results were entered by the treating doctors. Patient data from each IHC clinic was entered to an electronic database of the NAP and The Union on a regular basis. Every month, these IHC electronic data were sent to a central electronic database at the Union office in Mandalay, which was maintained by a Monitoring, Evaluation, Accountability and Learning Unit. Data quality control of all electronic databases was performed at regular intervals to minimize errors with data entry and to ensure data validity and consistency.

### 2.3. Study Population

The study included all adult and pediatric PLHIV who newly started first-line ART between 1 January 2016 and 31 December 2017 in all IHC clinics, Myanmar. The study group comprised of two cohorts: PLHIV initiated on ART between 1 January 2016 and 30 June 2017 who had first VL testing scheduled at 12-months and PLHIV initiated on ART between 1 July 2017 and 31 December 2017 who had first VL testing scheduled at 6-months.

### 2.4. Data Variables, Sources of Data and Data Collection Instrument

Data variables: these included patient ID number, date of starting ART, type of first-line ART regimen, age, sex, marital status, mode of HIV transmission, employment, literacy status, baseline WHO clinical stage, baseline CD4-cell count, co-infection with Hepatitis B (HBsAg positive) and Hepatitis C (HCV-antibody positive), diagnosis of TB at start of ART, programmatic outcome at time of VL testing at either 12-months or 6-months (retained on ART, died, loss to follow-up, stopped treatment or transferred-out), first VL test done, date of first VL test, VL > 1000 copies/mL, repeat VL test done, date of the repeat VL test, repeat VL > 1000 copies/mL, switched to second-line ART and date of switching to second-line ART. Data for the VL cascade and outcomes were censored on March 31, 2019.

Operational definitions: a first viral load test done at 12-months was programmatically defined as PLHIV having a VL test done between 10 and 15 months after starting ART. A first viral load test done at 6-months was programmatically defined as PLHIV having a VL test between 4 and 9 months after starting ART. In each case the time range was there to allow for PLHIV having difficulties in accessing clinics. VL tests done outside of these times were defined as unscheduled tests.

Source of data on patient characteristics and laboratory results including VL was the central electronic database of the NAP and The Union in Mandalay.

Data collection: Data were collected by exporting the data in the central electronic data base to an EXCEL file in April 2019.

### 2.5. Analysis and Statistics

Data were exported from the EXCEL file into STATA and analyzed using STATA (version 12.1 STATA Corp., College Station, TX, USA). Frequencies and proportions were used to summarize categorical variables and medians and interquartile ranges (IQR) used to summarize continuous variables. There were two cohorts of PLHIV: (i) those with first VL testing at 12-months after initiating ART and (ii) those with first VL testing at 6-months after initiating ART. Characteristics of PLHIV, viral load testing and outcomes between the two cohorts were compared using Pearson’s chi-square test and chi square test for trend. Baseline risk factors associated with the diagnosis of virologic failure (VL > 1000 copies/mL on repeat VL testing) in those who were tested for VL at 12-months or 6-months were analyzed using risk ratios (RR) and 95% confidence intervals (CI). Factors that showed associations with *p*-value < 0.2 in bivariate analysis were included in an adjusted logistic regression model. Levels of significance were set at 5% (*p*-value < 0.05).

### 2.6. Study Permission and Ethics Approval

Permission for the study was obtained from the National AIDS Program (NAP). Ethics approval was obtained from the Ethical Review Committee, Department of Medical Research, Myanmar (Ethics/DMR/2018/131) and the Ethics Advisory Group, International Union Against Tuberculosis and Lung Disease, Paris, France (EAG number: 46/18). Data were collected in a designed format (unique code, age and sex) with no patient names and confidentiality was maintained by keeping patients’ files in a lockable cabinet and electronic data securely in a password protected computer. As secondary data were used, the need for informed patient consent was waived.

## 3. Results

### 3.1. Characteristics of the Two PLHIV Cohorts

There were 9129 PLHIV initiated on ART between 1 January 2016 and 31 December 2017: these included 7153 PLHIV who had VL testing scheduled for 12-months and 1976 who had VL testing scheduled for 6-months. Demographic and clinical characteristics of PLHIV in the two cohorts are shown in Table 1. The two cohorts were similar apart from a higher proportion of PLHIV who were HCV-antibody positive in the cohort to be VL tested at 6-months.

### 3.2. VL Testing in the Two PLHIV Cohorts

VL testing at and outside of the scheduled times is shown in Table 2. While the same percentage of PLHIV had VL testing done at the scheduled time of 12-months or 6-months, the 6-month cohort had significantly more VL tests done outside the scheduled time and significantly more VL tests done in total.

### 3.3. VL Testing Cascade and Outcomes

The cascade of VL testing in the 12-month cohort is shown in Figure 1. There were 6816 PLHIV who were retained and taking ART at 12-months, and of these 3476 (51%) had VL tested at the scheduled time. The characteristics of those VL tested at the scheduled time and those not VL tested were similar apart from some differences in the regions from where the PLHIV came from (data not shown). Of those having VL testing at the scheduled time, 10% had VL > 1000 copies/mL and 79% of them had a repeat VL test. Of these, 42% had VL > 1000 copies/mL and 85% of them were switched to second-line ART. For those tested at 12-months, the median time (IQR) between first and repeat VL tests was 112 (91–154) days and the median (IQR) time between the repeat VL test and switching to second-line ART was 56 (39–71) days. There were 1255 PLHIV (18% of the entire cohort) who had VL testing done outside of the scheduled time. The viral load testing cascade followed a similar pattern to those tested as scheduled, except fewer PLHIV had a repeat VL test after the first VL > 1000 copies/mL.

The cascade of VL testing in the 6-month cohort is shown in Figure 2. There were 1892 PLHIV who were retained and taking ART at 6-months and of these 952 (50%) had VL tested at the scheduled time. The characteristics of those VL tested at the scheduled time and those not VL tested were again similar apart from some differences in the regions from where the PLHIV came from (data not shown). Of those having VL testing at the scheduled time, 11% had VL > 1000 copies/mL and 83% of them had a repeat VL test. Of these, 26% had VL > 1000 copies/mL and 39% of them were switched to second-line ART. For those tested at 6-months, the median time (IQR) between first and repeat VL tests was 115 (91–146) days and the median (IQR) time between the repeat VL test and switching to second-line ART was 102 (72–156) days. There were 421 PLHIV (21% of the entire cohort) who had VL testing done outside of the scheduled time. There were some differences compared with those tested at the scheduled time: fewer PLHIV had a repeat VL test after their first VL > 1000 copies/mL but more PLHIV had a repeat VL > 1000 copies/mL and were switched to second-line ART.

A comparison of the VL testing cascade and outcomes in the 6-months and 12-months cohorts is shown in Table 3. The testing and outcomes of the two cohorts were similar up to the repeat VL test, but a higher proportion of PLHIV in the 12-month cohort had repeat VL > 1000 copies/mL and were switched to second-line ART. Altogether, 98 (85%) of 115 PLHIV in the 12-month cohort who had repeat VL > 1000 copies/mL were switched to second-line ART, which was significantly higher than 9 (36%) of 23 PLHIV in the 6-month cohort who had repeat VL > 1000 copies/mL, *p* < 0.001. Of the 17 PLHIV in the 12-month cohort with virologic failure who did not switch, 11 (65%) died or were lost to follow-up or transferred-out, which was similar to 6 (43%) of 14 PLHIV in the 6-month cohort who died or were lost to follow-up (see Table 3).

### 3.4. Risk Factors for Virologic Failure in those Having VL Testing

Baseline risk factors for virologic failure in those who were VL tested at 12-months are shown in Table 4. After adjusting for confounders, being married or widowed were protective against having virologic failure while being in WHO clinical stage 4 and having a CD4 cell count < 200 cells/µL were risk factors for virologic failure. Baseline risk factors for virologic failure in those who were VL tested at 6-months are shown in Table 5. After adjusting for confounders, having a CD4 cell count < 200 cells/µL was the only risk factor for virologic failure.

In both the 12-month and 6-month cohorts we conducted a sensitivity analysis where we classified PLHIV who had VL > 1000 and did not have a second VL test as having virologic failure (data not shown). There were no substantial changes in the main conclusions presented above for either of the two cohorts.

## 4. Discussion

There were four main findings from this observational cohort study on programmatic performance and outcomes of routine VL monitoring done at 12-months and at 6-months in PLHIV newly initiating ART in the IHC program in Myanmar.

First, baseline demographic, clinical and immunological characteristics of both cohorts were similar with about one third of PLHIV starting ART either in WHO clinical stage 3 and 4 or with CD4 cell counts< 200 cells/µL. Before 2017, national guidelines specified that PLHIV should start ART if they were in WHO clinical stage 3 or 4 or if the CD4 count < 500 cells/µL [18]. However, in 2017, the guidelines changed to align with those of the WHO [6], specifying that all PLHIV start ART regardless of the WHO clinical stage or CD4 cell count [19]. In the 6-month VL testing cohort who started ART from July 2017 onwards, we might therefore have expected to see more PLHIV in WHO clinical stage 1 and fewer in WHO clinical stage 4. However, this was not the case. Despite new guidance, failure to get PLHIV started early on ART continues to be a global problem [20], and not just unique to Myanmar, and concerted action will be needed at all levels to encourage more widespread HIV testing and adoption of the “Test and Treat” approach.

Second, in both cohorts about half of PLHIV had VL testing done at the scheduled time of 12-months or 6-months according to our definition and according to policy. Of those who had VL testing done outside the scheduled time, this was more common amongst the 6-month VL testing cohort. We did not collect detailed information on when these PLHIV had non-scheduled VL tests or the reasons for this non-compliance with policy. However, we speculate that this may have been due to clinicians requesting earlier VL testing based on the need to know, for example in pregnant women, or later VL testing based on patient clinical status. While previous studies have shown that routine VL monitoring results in earlier detection of virologic failure and earlier switching to second-line ART [21,22], a recent study in Vietnam showed no difference in rates of failure or death between routine and targeted VL testing [23].

Third, while only half of both cohorts had their first VL test done at the scheduled time, about 80% of PLHIV with VL > 1000 copies/mL on the first test had repeat VL tests done. This performance compares favorably with similar assessments from other countries [24]. Moreover, the time between first and repeat VL tests in both cohorts was less than 4 months, which is considerably faster than the 12 months recently reported from Rwanda [25]. Performance was better in the 12-month cohort where 85% of those with VL > 1000 copies/mL were switched to second-line ART. In contrast, in the 6-month cohort, less than 40% of those with VL > 1000 copies/mL were switched to second-line ART. Death and loss to follow-up were similar at this stage between both cohorts and in the remainder, we do not know the reasons why switching to second-line ART was less frequent in the 6-month cohort. The follow-up period of observation was shorter with the 6-month cohort and this may have just been a data collection issue. Clinicians, however, may have been reluctant to change to more demanding protease-inhibitor based ART so early on in the course of therapy and may have decided to wait. This may explain why it took twice as long to switch to second-line ART in the 6-month cohort (median 102 days) compared with the 12-month cohort (median 56 days). The decision to switch therapies is not an easy one. In a systematic review in 16 African countries, less than 60% of PLHIV with diagnosed virologic failure switched to second-line ART [26], with documented reasons being a desire by clinicians to optimize adherence counseling, no apparent evidence of PLHIV having failed on clinical or immunological grounds, perceived challenges with protease inhibitor-based therapy or concerns about patient compliance based on a past history of missing scheduled appointments [27,28].

Fourth, risk factors for virologic failure in both cohorts were advanced HIV-related disease and/or immunosuppression. These findings are in line with previous reports [29,30,31], although associations between advanced HIV disease and virologic failure are not always consistent [32,33]. Being married or widowed offered protection in the 12-month cohort against virologic failure and this is possibly due to the supportive efforts of spouses/families in ensuring adherence to treatment and compliance with follow-up.

The strengths of this study were the large numbers of patients initiated on ART in both cohorts, rigorous monitoring of hard milestones and end points along the VL testing cascade and the conduct and reporting of the study in line with the strengthening the reporting of observational studies in epidemiology (STROBE) guidelines [34].

There were, however, some limitations. First, we did not collect detailed information on PLHIV who had VL testing done outside of the scheduled times. However, the main purpose of our study was to assess VL testing and VL results for those who were tested at our defined 12-month and 6-month schedules, and we were less interested in those who were tested outside of these schedules for whom the interpretation of results would have been difficult. Second, there is the possibility of a selection bias or survival bias as we only included PLHIV who were VL tested in the risk factor analysis for VL failure. To address the selection bias, we compared the characteristics between patients included in the analysis and those who were not included and found that they were similar apart from some small differences in the regions from where they came from. Thus, we concluded that the potential impact of selected bias was small. To address survival bias and the possibility that we might have misclassified those with VL failure, we conducted a sensitivity analysis by classifying those who did not have VL testing a second time as VL failure and found that there was no substantial change in results and our main conclusion stays the same. Third, we did not have information on what happened to PLHIV who did not have their first or repeat VL tests done and we did not enquire as to why those with defined virologic failure were not switched to second-line ART. Fourth, we did not record details about when and how well EAC was performed between first and repeat VL tests. Fifth, the two cohorts (the 12-month VL testing cohort and the 6-month VL testing cohort) were enrolled and managed on ART at different but consecutive time periods. It is possible that the health system and health service factors were different in 2018 compared with 2017 and 2016, although we have no evidence to suggest that this was the case. Finally, the follow-up period for repeat VL testing and switching to second-line ART was shorter for the 6-month cohort and the lack of time for these events to have happened compared with the 12-month cohort may be one of the explanations for a low proportion of the 6-month testing cohort switching to second-line ART.

Despite these limitations, there are some important programmatic implications from this study. First, VL testing is key to monitoring VL suppression and has to be done to assess progress against the UNAIDS 90-90-90 targets [2]. The policy is now to VL test routinely at 6-months and concerted efforts are needed to move implementation from 50% of PLHIV being tested to as near 100% as possible. This will require a three-pronged approach, focusing on close monitoring and supervision, health facility strengthening and improving laboratory systems. Regular structured supervision improves and sustains ART program performance, as has been shown elsewhere [35]. A quality improvement program focused on improving knowledge, skills and responsibilities of staff at health facilities significantly increased VL testing performance in Malawi [36], and this approach could be considered in Myanmar. VL testing also needs to be decentralized to overcome the logistic hurdles associated with transporting blood specimens to central laboratories. The Xpert HIV-1 VL assay is a new automated molecular test that utilizes the GeneXpert platform system widely used for tuberculosis diagnosis. This performs well compared to current reference tests [37] and Markov models suggest that this potential point-of-care VL assay could be decentralized and be cost-effective [38].

Second, once the first VL test is done close attention must be paid to following through the VL cascade. Those who have VL > 1000 copies/mL must be referred early for good quality EAC under the guidance of medical officers and a repeat VL test must be done about 3 months later. Both primary and acquired ARV drug resistance are becoming a global problem [8,9], and rapid identification and change to alternative ART is needed in those with proven ARV resistance to avoid amplification and spread of further resistance. In order to better understand what VL > 1000 copies/mL means on repeat testing drug resistance studies should be carried out to assess the prevalence and patterns of ARV drug mutations.

Finally, how clinicians and PLHIV view the early timing of VL testing and the decision to switch to second-line ART may change as new WHO guidelines become adopted. Since this study was done, the WHO has recommended changes to first-line ART, such as the use of dolutegravir (DTG) to replace efavirenz [39]. DTG, an integrase inhibitor, is effective and shows good tolerability, limited drug–drug interactions and a high barrier to resistance and this may be associated with more rapid viral suppression during routine testing. The use of DTG in first-line regimens, which is being rapidly adopted in many low- and middle-income countries, may therefore encourage clinicians to use more 6-month VL testing and to change more rapidly to second-line ART if VL failure is diagnosed. For countries that have not yet moved to DTG-based first-line regimens, the WHO also recommends DTG in combination with an optimized nucleoside reverse-transcriptase inhibitor backbone as the preferred second-line regimen [39]. This is much easier for patients to take than the previous protease-inhibitor based second line ART and again may encourage more VL testing in the future.

## 5. Conclusions

This study showed that about half of PLHIV initiated on ART had VL testing at 12-months or 6-months in line with policy, with an additional one fifth having the tests done outside of the scheduled times. In terms of the VL cascade, a similar and satisfactory pattern was seen in both cohorts up to VL retesting, but thereafter less PLHIV in the 6-month cohort were identified with virologic failure and fewer were switched to second-line ART compared with the 12-month cohort. Reasons for these findings and programmatic implications are discussed.

## Figures and Tables

**Figure 1 tropicalmed-05-00140-f001:**
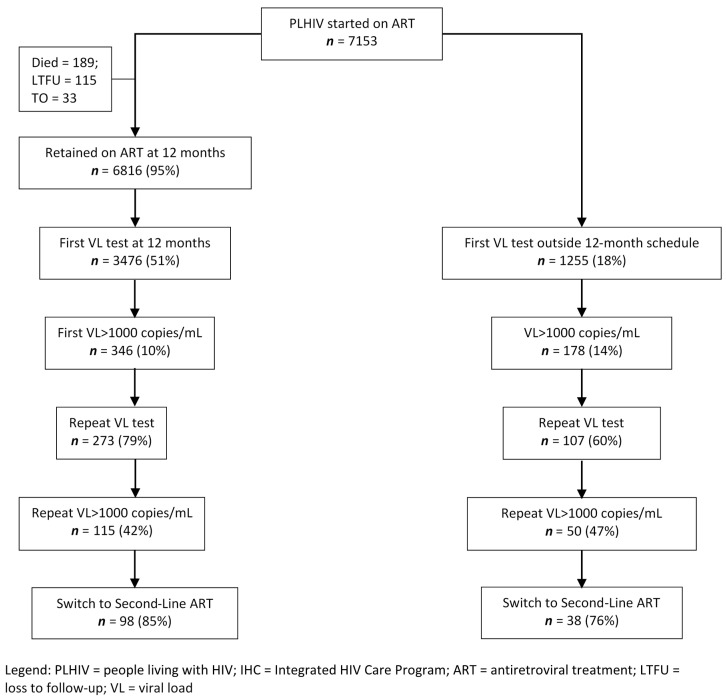
Flow diagram of PLHIV initiating first-line ART in the IHC program, Myanmar, between 1 January 2016 and 30 June 2017 who were scheduled for viral load testing at 12-months and the outcomes of testing.

**Figure 2 tropicalmed-05-00140-f002:**
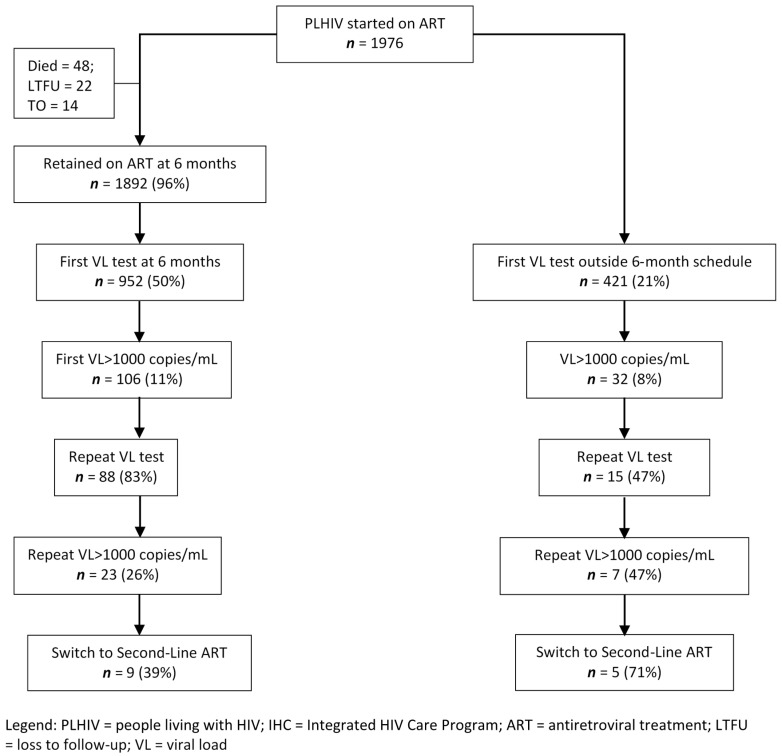
Flow diagram of PLHIV initiating first-line ART in the IHC program, Myanmar, between 1 July 2017 and 31 December 2017 who were scheduled for viral load testing at 6-months and the outcomes of testing.

**Table 1 tropicalmed-05-00140-t001:** Characteristics of people living with HIV (PLHIV) starting first-line antiretroviral therapy (ART) in the integrated HIV care (IHC) program in Myanmar: the cohort enrolled between 1 January 2016 and 30 June 2017 with viral load scheduled at 12-months and the cohort enrolled between 1 July 2017 and 31 December 2017 with viral load scheduled at 6-months.

Baseline Characteristics at Time of Starting ART		VL Testing Scheduled at 12-Months		VL Testing Scheduled at 6-Months	
		*n*	(%)	*n*	(%)
Total		7153		1976	
Age group in years:	0–14	544	(7.6)	130	(6.6)
	15–59	6462	(90.3)	1794	(90.8)
	60 and above	147	(2.1)	52	(2.6)
Gender:	Male	4245	(59.3)	1212	(61.3)
	Female	2908	(40.7)	764	(38.7)
Marital status:	Single	1838	(25.7)	516	(26.1)
	Married	3855	(53.9)	1048	(53.0)
	Divorced	481	(6.7)	127	(6.4)
	Widowed	795	(11.1)	221	(11.2)
	No data	184	(2.6)	64	(3.2)
Mode of HIV transmission:	Heterosexual	4887	(68.3)	1413	(71.5)
	MSM	115	(1.6)	37	(1.9)
	Sex worker	31	(<1)	6	(<1)
	PWID	680	(9.5)	177	(9.0)
	MTCT	559	(7.8)	127	(6.4)
	Blood TF	190	(2.7)	38	(1.9)
	No data	691	(9.7)	178	(9.0)
Employment status:	Employed	5133	(71.8)	1381	(70.0)
	No data	250	(3.5)	97	(4.9)
Literacy status:	Literate	6088	(85.1)	1720	(87.0)
	No data	166	(2.3)	65	(3.3)
WHO clinical stage:	WHO stage 1	1730	(24.2)	506	(25.6)
	WHO stage 2	1293	(18.1)	330	(16.7)
	WHO stage 3	1844	(25.8)	508	(25.7)
	WHO stage 4	418	(5.8)	110	(5.6)
	No data	1868	(26.1)	522	(26.4)
CD4 cell count–cells/µL:	≥200	2428	(33.9)	620	(31.4)
	<200	2305	(32.2)	641	(32.4)
	No data	2420	(33.8)	715	(36.2)
Hepatitis B co-infection:	Yes	517	(7.2)	137	(6.9)
	No data	545	(7.6)	243	(12.3)
Hepatitis C co-infection:	Yes	720	(10.1)	220	(11.1) ^a^
	No data	546	(7.6)	243	(12.3)
TB at start of ART:	Yes	861	(12.0)	226	(11.4)
	No data	1874	(26.2)	521	(26.4)
First-line ART regimen:	TDF-based	6319	(88.3)	1773	(89.7)
	AZT-based	356	(5.0)	88	(4.5)
	ABC-based	472	(6.6)	115	(5.8)
	D4T-based	6	(<1)	0	-

PLHIV = people living with HIV; ART = antiretroviral therapy; IHC = integrated HIV care program; VL = viral load; MSM = men who have sex with men; PWID = people who inject drugs; MTCT = mother to child transmission of HIV; TF = transfusion; WHO = World Health Organization; TDF = tenofovir; AZT = zidovidine; ABC = abacavir; D4T = stavudine. ^a^
*p* < 0.05 compared with the cohort that was VL tested at 12-months.

**Table 2 tropicalmed-05-00140-t002:** Viral load testing at and outside of scheduled times of 12-months or 6-months in PLHIV initiating ART in the IHC program in Myanmar.

Characteristics	VL Testing Scheduled at 12-Months		VL Testing Scheduled at 6-Months		*p*-Value
	*n*	(%)	*n*	(%)	
PLHIV starting ART	7153		1976		
First VL test done at scheduled time	3476	(48.6)	952	(48.2)	0.74
First VL test done outside scheduled time	1255	(17.5)	421	(21.3)	<0.001
Total first VL tests done	4731	(66.1)	1373	(69.5)	<0.01

PLHIV = people living with HIV; ART = antiretroviral therapy; IHC = integrated HIV care program; VL = viral load. In each column, the denominators for all the percentages are all PLHIV who were newly initiated on ART. The percentages do not take account for those who were lost to follow-up, who died or were transferred-out by 12-months or by 6-months.

**Table 3 tropicalmed-05-00140-t003:** Viral load testing and outcomes in PLHIV initiating ART in the IHC program in Myanmar in relation to whether VL was done at the scheduled 12-months or the scheduled 6-months.

Characteristics	VL Testing Scheduled at 12-Months		VL Testing Scheduled at 6-Months		*p*-Value
	*n*	(%)	*n*	(%)	
PLHIV starting ART	7153		1976		
PLHIV retained at 12 or 6 months	6816	(95)	1892	(96)	0.39
First VL test done at the scheduled time	3476	(51)	952	(50)	0.60
First VL > 1000 copies/mL	346	(10)	106	(11)	0.29
Repeat VL test	273	(79)	88	(83)	0.36
Repeat VL > 1000 copies/mL	115	(42)	23	(26)	<0.01
Switch to second-line ART	98 ^a^	(85)	9 ^b^	(39)	<0.001

PLHIV = people living with HIV; ART = antiretroviral therapy; IHC = integrated HIV care program; VL = viral load. Percentages in each row are derived from the row above. ^a^ There were 17 PLHIV not switched to second-line ART. Of these, there were 11 with program attrition: 2 died, 7 were lost to follow-up and 2 transferred-out. There were 6 in whom reasons for not switching were not known. ^b^ There were 14 PLHIV not switched to second-line ART. Of these, there were 6 with program attrition: 2 died and 4 were lost to follow-up. There were 8 in whom reasons for not switching were not known.

**Table 4 tropicalmed-05-00140-t004:** Risk factors for PLHIV on first-line ART in the IHC program in Myanmar who were viral load tested at 12-months and who were diagnosed with virologic failure.

Risk Factors at Start of ART		VL at 12-Months	Diagnosisof VF		RR (95% CI)	aRR (95% CI)
		*n*	*n*	(%)		
Total		3476	115	(3.3)		
Age group in years:	0–14	270	12	(4.4)	Ref	Ref
	15–59	3136	101	(3.2)	0.7(0.4–1.3)	1.5(0.5–4.3)
	60 and above	70	2	(2.9)	0.6(0.1–2.8)	1.3(0.3–7.1)
Gender:	Male	2065	68	(3.3)	Ref	Ref
	Female	1411	47	(3.3)	1.0(0.7–1.5)	1.2(0.8–1.8)
Marital status:	Single	940	43	(4.6)	Ref	Ref
	Married	1855	53	(2.9)	0.6(0.4–0.9)	0.6(0.4–0.9)
	Divorced	225	9	(4.0)	0.9(0.4–1.8)	0.9(0.4–1.8)
	Widowed	390	9	(2.3)	0.5(0.2–1.0)	0.4(0.2–0.9)
	No data	66	1	(1.5)		
Employment status:	Employed	2504	77	(3.1)	Ref	Ref
	Unemployed	877	35	(4.0)	1.3(0.9–1.9)	1.2(0.7–1.9)
	No data	95	3	(3.2)		
Literacy status:	Literate	3001	99	(3.3)	Ref	Ref
	Illiterate	402	14	(3.9)	1.1(0.6–1.8)	0.9(0.5–1.7)
	No data	73	2	(2.7)		
WHO clinical stage:	WHO stage 1	894	21	(2.4)	Ref	Ref
	WHO stage 2	677	26	(3.8)	1.6(0.9–2.9)	1.6(0.9–2.8)
	WHO stage 3	853	33	(3.9)	1.6(0.9–2.8)	1.6(0.9–2.9)
	WHO stage 4	211	16	(7.6)	**3.2(1.7–6.1)**	**3.4(1.7–6.7)**
	No data	841	19	(2.3)		
CD4 cells/µL:	≥200	1218	24	(1.9)	Ref	Ref
	<200	1065	43	(4.0)	**1.9(1.2–3.1)**	**1.9(1.2–3.2)**
	No data	1193	48	(4.0)		
TB at start of ART:	No	2241	82	(3.7)	Ref	Ref
	Yes	397	15	(3.8)	1.0(0.6–1.8)	0.6(0.3–1.1)
	No data	838	18	(2.2)		
First-line ART	TDF-based	3108	98	(3.2)	Ref	Ref
	AZT-based	170	6	(3.5)	1.1(0.5–2.5)	1.2(0.5–3.3)
	ABC-based	197	11	(5.6)	1.8(0.9–3.2)	1.8(0.7–4.6)
	D4T-based	1	0			

PLHIV = people living with HIV; ART = antiretroviral therapy; IHC = integrated HIV care program; VL = viral load; VF = virologic failure; RR = relative risk; aRR = adjusted relative risk; CI – confidence intervals; WHO = World Health Organization; TDF = tenofovir; AZT = zidovidine; ABC = abacavir; D4T = stavudine. Mode of HIV transmission, Hepatitis B and Hepatitis C status showed no associations with VF and were not included in the table; the *p* values were >.0.2 and the data are not shown. Bolded RR and 95% CI indicate a statistically significant result at *p* < 0.05 level.

**Table 5 tropicalmed-05-00140-t005:** Risk factors for PLHIV on first-line ART in the IHC program in Myanmar who were viral load tested at 6-months and who were diagnosed with virologic failure.

Risk Factors at Start of ART		VL at 6-Months	Diagnosis of VF		RR (95% CI)	aRR (95% CI)
		*n*	*n*	(%)		
Total		952	23	(2.4)		
Age group in years:	0–14	57	2	(3.5)	Ref	Ref
	15–59	873	20	(2.3)	0.7(0.2–2.7)	0.4(0.1–5.9)
	60 and above	22	1	(4.6)	1.3(0.1–13)	1.9(0.1–36)
Gender:	Male	577	16	(2.8)	Ref	Ref
	Female	375	7	(1.9)	0.6(0.3–1.6)	0.9(0.4–2.6)
Marital status:	Single	240	6	(2.5)	Ref	Ref
	Married	535	15	(2.8)	1.1(0.5–2.9)	1.3(0.5–3.7)
	Divorced	63	0	(0)		
	Widowed	88	1	(1.1)	0.5(0.1–3.7)	0.3(0.1–2.6)
	No data	26	1	(3.9)		
Employment status:	Employed	686	17	(2.5)	Ref	Ref
	Unemployed	224	3	(1.3)	0.5(0.2–1.8)	0.4(0.1–1.7)
	No data	42	3	(7.1)		
Literacy status:	Literate	865	21	(2.4)	Ref	Ref
	Illiterate	63	1	(1.6)	0.7(0.1–4.8)	1.2(0.1–11)
	No data	24	1	(4.2)		
WHO clinical stage:	WHO stage 1	247	4	(1.6)	Ref	Ref
	WHO stage 2	189	4	(2.1)	1.3(0.3–5.1)	0.9(0.2–3.7)
	WHO stage 3	231	11	(4.8)	2.9(0.9–9.1)	1.3(0.3–4.8)
	WHO stage 4	51	1	(2.0)	1.2(0.1–10)	0.4(0.1–3.9)
	No data	234	3	(1.3)		
CD4 cells/µL:	≥200	321	2	(0.6)	Ref	Ref
	<200	290	13	(4.5)	**7.2(1.6–31)**	**7.8(1.7–35)**
	No data	341	8	(2.4)		
TB at start of ART:	No	609	13	(2.1)	Ref	Ref
	Yes	109	7	(6.4)	3.0(1.2–7.4)	2.7(0.9–8.5)
	No data	234	3	(1.3)		
First-line ART:	TDF-based	868	20	(2.3)	Ref	Ref
	AZT-based	39	1	(2.6)	1.1(0.2–8.1)	0.8(0.1–8.8)
	ABC-based	45	2	(4.4)	1.9(0.5–7.9)	2.1(0.2–18)

PLHIV = people living with HIV; ART = antiretroviral therapy; IHC = integrated HIV care program; VL = viral load; VF = virologic failure; RR = relative risk; aRR = adjusted relative risk; CI – confidence intervals; WHO = World Health Organization; TDF = tenofovir; AZT = zidovidine; ABC = abacavir; D4T = stavudine. Mode of HIV transmission, Hepatitis B and Hepatitis C status showed no associations with VF and were not included in the table; the *p* values were >.0.2 and the data are not shown. Bolded RR and 95% CI indicate a statistically significant result at *p* < 0.05 level.

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
