# Peer review of "Performance and Outcomes of Routine Viral Load Testing in People Living with HIV Newly Initiating ART in the Integrated HIV Care Program in Myanmar between January 2016 and December 2017"

_tropicalmed, 2020, doi:10.3390/tropicalmed5030140_

Round 1
Reviewer 1 Report
Well-written manuscript evaluating the implementation of routine viral load monitoring in Myanmar.
The discussion is lengthy and should be shortened.
On page 5 line 319 it is stated: Third, the VL testing cascade worked will in both cohorts up to the repeat VL tests...
With only half in both having VL testing done at planned time, I think the sentence should be re-phrased.
It would be interesting if those having vs those not having VL testing done at 6/12 months could be compared. Do the population not having VL testing done differ from those being tested on time?
In Table 1 The heading mode of HIV transmission has been misplaced. I supposed it should be pushed 3 lines upwards.
Author Response
We thank Reviewer 1 for taking the time and effort to read and comment on our paper. We respond to each of the comments below. We have made changes to the manuscript using red font.
Well-written manuscript evaluating the implementation of routine viral load monitoring in Myanmar.
Response:
Thank you
The discussion is lengthy and should be shortened.
Response:
We agree that the Discussion is quite long but we felt we needed to explain in some detail the key findings, especially in relation to previous literature, we needed to be transparent about our limitations and we had a number of programmatic implications that needed explanation as well. We have tried to cut down words here and there but have not been able to drastically shorten the section. We hope the reviewer and editor can accept our stance here.
On page 5 line 319 it is stated: Third, the VL testing cascade worked will in both cohorts up to the repeat VL tests: “With only half in both having VL testing done at planned time”, I think the sentence should be re-phrased.
Response
Thank you. We have rephrased this on line 319 as follows: “Third, while only half of both cohorts had their first VL test done at the scheduled time, about 80% of PLHIV with VL>1000 copies/mL on the first test had repeat VL tests done.”
It would be interesting if those having vs those not having VL testing done at 6/12 months could be compared. Do the population not having VL testing done differ from those being tested on time?
Response:
Thank you. We did in fact include statements to this effect in the Results narrative. On lines 219-221 we stated for the 12-month testing cohort: “The characteristics of those VL tested at the scheduled time and those not VL tested were similar apart from some differences in the regions from where the PLHIV came from (data not shown).” On lines 235-236 we stated for the 6-month testing cohort: “The characteristics of those VL tested at the scheduled time and those not VL tested were again similar apart from some differences in the regions from where the PLHIV came from (data not shown).”
In Table 1 The heading mode of HIV transmission has been misplaced. I supposed it should be pushed 3 lines upwards.
Response:
Thank you for pointing this out. It is now in line with the first row indicating PWID.
Reviewer 2 Report
This manuscript is a significant contribution to the field, and I recommend it to be accepted pending suggested changes.
- Note: I have used bold to add new suggested text in some areas and strikethrough for suggested deletions.
Comment#1:
Lines 139 - 140: “Every patient registered in the IHC program has had a dedicated file in which baseline and follow-up clinical details and laboratory test results are were entered by the treating doctors.”
Overall, please make sure that activities performed during the study are described in the past tense throughout the manuscript.
Comment#2:
Lines 175-176: The following sentence is not very clear “There were two cohorts of PLHIV: those VL tested at 12-months and those VL tested at 6-months.”
- Suggestion: “There were two cohorts of PLHIV: 1) those with first VL testing at 12-months after initiating ART and 2) those with first VL testing at 6-months after initiating ART.”
Comment #3:
The percentages listed in figure 1 do not match those listed in table 2 for the same items – please review and adjust the percentages to have a match between those 2.
For example, for “first VL test at 12 months”; n = 3476 (51%) in figure 1; and, n = 3476 (48.6%) in table 2
Comment #4:
We have the same issue of discrepancies between figure 1 and the text: For example, in line 222 – it says “47% of those with a repeat VL test had VL>100 copies/mL; while that same percentage is 42% in figure 1.
- Overall, make sure the percentages listed in the text, the figures and the tables do match.
Comment#5:
For the sentence “47% of those with a repeat VL test had VL>100 copies/ML”; you cannot start a sentence with an Arabic number.
The sentence can be rewritten as “Forty seven percent of those ….”
Comment#6:
Lines 323 - 324: “Thereafter, performance was better in the 12-month cohort where 85% of those with VL>1000 copies/mL were switched to second-line ART.
“Thereafter” does not fit well in this sentence – please change it for a better transitional word.
It also says that performance was better in the “12-month cohort”; this is confusing – Looks like this is the opposite of your conclusions showing that the 6-months cohort did better than the 12-months cohort
Comment#7:
Line 356 – 357: The sentence “Hence, we judge the impact of selection bias to be small.” is not well rewritten; and the word “hence” is not in its right place here.
The phrase could be rewritten as: “Thus, we concluded that the potential impact of selection bias is expected to be small.”
Comment#8:
Lines 373 – 374: The sentence “The policy is now to VL test routinely at 6-months and concerted efforts are needed to move implementation from 50% to as near 100% as possible.” is not clear – please, rewrite for more clarity – for example 50% of what? 50% uptake?
Comment#9:
In your discussion, please say something about adherence assessment for those with a VL measure over 1000 copies/mL.
In addition to resistance testing; adherence measures and related counseling should be taken into account in the management of patients with virological failure.
If this was not done it should be mentioned as a weakness in your discussion
Author Response
Reviewer 2:
We thank Reviewer 2 for taking the time and effort to read and comment on our paper. We respond to each of the comments below. We have made changes to the manuscript using red font.
This manuscript is a significant contribution to the field, and I recommend it to be accepted pending suggested changes. Note: I have used bold to add new suggested text in some areas and strikethrough for suggested deletions.
Response
Thank you for this encouraging overall comment.
Comment#1:
Lines 139 - 140: “Every patient registered in the IHC program has had a dedicated file in which baseline and follow-up clinical details and laboratory test results are were entered by the treating doctors. Overall, please make sure that activities performed during the study are described in the past tense throughout the manuscript.
Response:
Thank you for this good suggestion. We have changed to the past tense all the narrative from lines 113 to 145.
Comment#2:
Lines 175-176: The following sentence is not very clear “There were two cohorts of PLHIV: those VL tested at 12-months and those VL tested at 6-months.” Suggestion: “There were two cohorts of PLHIV: 1) those with first VL testing at 12-months after initiating ART and 2) those with first VL testing at 6-months after initiating ART.”
Response:
Thank you. We have changed the sentence on lines 175-176 exactly as you suggest.
Comment #3:
The percentages listed in figure 1 do not match those listed in table 2 for the same items – please review and adjust the percentages to have a match between those 2.
For example, for “first VL test at 12 months”; n = 3476 (51%) in figure 1; and, n = 3476 (48.6%) in table 2
Response:
Thank you. The denominators in Figure 1 and in Table 2 are different. In Figure 1 the denominator is the number of PLHIV retained on ART after 12-months after taking account of those lost to follow-up, those who died and those transferred out. In Table 2 the denominator is the number of PLHIV who were newly initiated on treatment. This was explained in the table footnote but we have now made this more transparent as follows:” In each column, the denominators for all the percentages are all PLHIV who were newly initiated on ART. The percentages do not take account for those who were lost to follow-up, who died or were transferred out by 12-months or by 6-months.” We felt it was important to show both denominators as programmes use both for their reporting.
Comment #4:
We have the same issue of discrepancies between figure 1 and the text: For example, in line 222 – it says “47% of those with a repeat VL test had VL>100 copies/mL; while that same percentage is 42% in figure 1. Overall, make sure the percentages listed in the text, the figures and the tables do match.
Response:
Thank you for noticing this. These are indeed errors and we apologise. We have corrected the narrative percentages on lines 223-224 and on line 239 so that they match what is shown in the two Figures.
Comment#5:
For the sentence “47% of those with a repeat VL test had VL>100 copies/ML”; you cannot start a sentence with an Arabic number. The sentence can be rewritten as “Forty seven percent of those ….”
Response.
Thank you. We have in fact rephrased line 223 as follows: “Of these, 42% had VL>1000 copies/mL….”
Comment#6:
Lines 323 - 324: “Thereafter, performance was better in the 12-month cohort where 85% of those with VL>1000 copies/mL were switched to second-line ART.
“Thereafter” does not fit well in this sentence – please change it for a better transitional word.
It also says that performance was better in the “12-month cohort”; this is confusing – Looks like this is the opposite of your conclusions showing that the 6-months cohort did better than the 12-months cohort.
Response:
Thank you for this observation. In line 324, we have removed the word “Thereafter, …” and we have rephrased the sentence in lines 326-328 as follows: “Death and loss to follow-up were similar at this stage between both cohorts and in the remainder, we do not know the reasons why switching to second-line ART was less frequent in the 6-month cohort.”
Comment#7:
Line 356 – 357: The sentence “Hence, we judge the impact of selection bias to be small.” is not well rewritten; and the word “hence” is not in its right place here.
The phrase could be rewritten as: “Thus, we concluded that the potential impact of selection bias is expected to be small.”
Response:
Thank you. We have removed the word “Hence” and have rephrased as follows: “Thus, we concluded that the potential impact of selection bias was small.”
Comment#8:
Lines 373 – 374: The sentence “The policy is now to VL test routinely at 6-months and concerted efforts are needed to move implementation from 50% to as near 100% as possible.” is not clear – please, rewrite for more clarity – for example 50% of what? 50% uptake?
Response:
Thank you. In line 375, we have rephrased this as “from 50% of PLHIV being tested to as near 100% as possible.”
Comment#9:
In your discussion, please say something about adherence assessment for those with a VL measure over 1000 copies/mL. In addition to resistance testing, adherence measures and related counselling should be taken into account in the management of patients with virological failure. If this was not done it should be mentioned as a weakness in your discussion.
Response: Thank you. In lines 388 -389, we have emphasised the need for enhanced adherence counselling (EAC) under the guidance of medical officers.